# Voluntary HIV Testing and Counselling Initiatives in Occupational Settings: A Scoping Review

**DOI:** 10.3390/ijerph22020263

**Published:** 2025-02-12

**Authors:** Holly Blake, Mehmet Yildirim, Stephanie J. Lax, Catrin Evans

**Affiliations:** 1School of Health Sciences, University of Nottingham, Nottingham NG7 2HA, UK; mehmet.yildirim1@nottingham.ac.uk (M.Y.); catrin.evans@nottingham.ac.uk (C.E.); 2NIHR Nottingham Biomedical Research Centre, Nottingham NG7 2UH, UK; 3Lifespan and Population Health, School of Medicine, University of Nottingham, Nottingham NG5 1PB, UK; stephanie.lax@nottingham.ac.uk

**Keywords:** human immunodeficiency virus, voluntary HIV testing and counselling, HIV testing, workplace, public health, occupational settings, scoping review

## Abstract

Voluntary HIV testing and counselling (VCT) in the workplace could reach population groups who may be at risk for HIV but may not readily seek out testing from other services. We conducted a scoping review to understand (a) the nature of evidence related to initiatives and interventions for vocationally active adults on VCT in occupational settings, and (b) any facilitators and barriers to the delivery of and/or engagement with VCT initiatives/interventions in the workplace. JBI scoping review methodology was followed. The protocol was pre-registered. Included studies focused on vocationally active adults (population), VCT interventions or initiatives (concept), and workplaces in any sector or country (context). The review included studies published after 2000, in English, and of any research design. Studies relating to mandatory workplace HIV screening were excluded. MEDLINE, CINAHL, Scopus, PsycINFO, and the Cochrane Central Register of Control Trials were searched. Sources of grey literature included Google Scholar and governmental and organisational websites. One reviewer screened titles and abstracts; a second reviewer independently screened 10%. Data extraction utilised a modified JBI data extraction tool. We identified 17 studies reporting on 12 workplace VCT interventions (20,985 participants, 15–70 years). Studies were conducted in eight countries between 2001 and 2022. Interventions were delivered in organisations of different types, sizes and sectors. Testing included rapid blood tests and oral fluid self-tests. Where reported, the average on-site HIV testing uptake rate was 63%, and the average linkage to care rate was 86.85%. Views of workers, employers and service providers were largely positive. Barriers included being male, masculinity-driven workplace culture, HIV-related stigma, poor knowledge, low risk perceptions, lack of time and low support. Facilitators included on-site testing for convenience and accessibility, rapid and free tests, organisational, managerial and peer support, and embedding HIV tests within general health checks. Evaluation methods varied, although randomised trial designs were uncommon. Despite the limited number of studies, the workplace appears to be a viable route to the delivery of community-based VCT, albeit barriers should be addressed. Reporting quality of interventions and associated evaluations is variable and could be improved with the use of appropriate checklists.

## 1. Introduction

The Human Immunodeficiency Virus (HIV) specifically targets and damages the body’s immune system, gradually deteriorating its function and leaving the infected individual vulnerable to life-threatening infections and diseases [1]. If left untreated, HIV can progress to acquired immunodeficiency syndrome (AIDS), which ultimately leads to death [2]. HIV primarily spreads through unprotected sexual intercourse, sharing needles or syringes, and from mother to child during childbirth or breastfeeding. The global impact of HIV/AIDS is significant; in 2023, there were an estimated 39.9 million people living with HIV, with 1.3 million new infections and 630,000 AIDS-related deaths reported in the same year [3]. These statistics highlight the pressing need for prevention, treatment, and support programmes on a global scale.

HIV testing plays a crucial role in the prevention of the spread of the virus. Global data from 2023 show that 14% of people living with HIV are not aware of their HIV status [3], which highlights the need for increased testing efforts. Early detection of HIV is, therefore, a life-saving step as it allows for prompt initiation of treatment, which not only improves individual health outcomes but also reduces the risk of transmission to others [4]. Although there is currently no cure for HIV, antiretroviral therapy (ART) effectively suppresses viral replication, leading to lower viral loads in the blood and reduced transmission risk [5]. The concept of ‘Undetectable = Untransmittable’ (U = U) is relevant here because it highlights that people living with HIV who are on treatment and have a fully suppressed viral load have a zero risk of transmitting the virus to their sexual partners. Efforts to maximise early detection through screening are, therefore, critical.

Based on employment-to-population ratios worldwide, around 58% of the global working-age population is employed [6]. Given the high proportion of time adults spend at work, the workplace is increasingly seen as an important avenue for promoting health and wellbeing [7,8]. Health screening is becoming a common feature in workplace health promotion programmes, such as for diabetes [9], cardiovascular disease risk [10], mental health [11], and general health checks [12]. However, workplace interventions rarely include HIV testing and counselling [13]. The inclusion of voluntary HIV testing and counselling (VCT) in the context of workplace health promotion may help to reduce HIV-related stigma and increase access to testing. Further, certain occupational settings pose a higher risk for HIV transmission among vocationally active adults. The risk for HIV transmission in workplace settings is increased by exposure to blood, body fluids, or tissues while undertaking job-related tasks. Examples include health workers, police officers, fire fighters, and correctional facility personnel. Also, settings that typically encompass male-dominated workplaces, such as the construction industry [14], may be linked to locations where there may be high levels of sex work. Risk is also high in work environments involving direct exposure to blood or other infectious substances, such as through unsafe injection equipment, like needles or syringes [15].

Existing knowledge in this area is limited, and there is a need to map out research on initiatives and interventions for VCT in the workplace in terms of ‘how’ and ‘where’ they have been implemented and evaluated. There is also a need to identify the facilitators and barriers that exist in relation to the delivery of, and participation in, VCT in workplace settings. Exploring these factors can provide a better understanding of the current efforts and challenges in promoting VCT among vocationally active adults in occupational settings. This knowledge could contribute to the development of targeted strategies and interventions to improve HIV prevention, awareness, and support in community contexts.

A preliminary search of MEDLINE, the Cochrane Database of Systematic Reviews, and JBI Evidence Synthesis was conducted in October 2023, and no current or underway systematic reviews or scoping reviews on the topic were identified. There were a limited number of reviews available focusing on preventing risky sexual behaviours [16], community-based approaches [17], and barriers to workplace HIV testing in South Africa [18], but they did not cover a global perspective or provide insights from an occupational standpoint on applied VCT. Therefore, we aimed to conduct a scoping review to describe and understand what research has been undertaken on initiatives and interventions about VCT in occupational settings, in a global context.

The review questions were:

1. What is the evidence related to initiatives and interventions for vocationally active adults on VCT in occupational settings?

2. What are the facilitators and barriers to the delivery of and/or engagement with VCT initiatives/interventions in the workplace?

## 2. Materials and Methods

The scoping review was conducted in accordance with the JBI methodology for scoping reviews [19] and reported in line with the Preferred Reporting Items for Systematic Reviews and Meta-Analyses extension for Scoping Reviews (PRISMA-ScR) [20] (Appendix A). The protocol was pre-registered: https://doi.org/10.17605/OSF.IO/BAMJ7 (accessed on 10 Feb 2025). A scoping review was appropriate as the aim was to describe, map and characterise the evidence rather than synthesise data to answer a focused clinical question [21].

### 2.1. Inclusion Criteria

The inclusion and exclusion criteria follow the participant, concept, and context (PCC) framework [22].

Participants: This review included studies involving vocationally active adults. Vocationally active adults are defined as people who are currently engaged in the workforce, pursuing a career, or engaging in vocational activities such as seasonal working. This term was deliberately chosen since it includes all types of working status involving being employed, self-employed, or actively seeking employment while utilising their skills, knowledge, and expertise in a specific field or occupation. The review focus intended to be on adult populations as defined in the United Kingdom (UK) (18 years of age or over), but studies with an age range of 15 years or over were included as it was not possible to separate the results. This decision was made because the initial search identified some studies that presented age ranges from 15 to 24 years.

Concept: This review included any study that explores or evaluates opt-in VCT initiatives or interventions. Mandatory HIV screening via occupational health surveillance programmes was excluded.

Context: Any type of occupational setting without geographical limitation was included in the review. However, studies relating to sex work were excluded due to the highly variable context and occupational/legal status of sex work settings worldwide (i.e., ranging from highly criminalised and not considered an ‘occupation’ nor governed by labour laws in some settings to a more formalised legal status in other settings) [23]. Due to the severe stigma and ambiguous legal/occupational status of sex work globally, it was felt that this context was not comparable to other potential settings and would require a separate review that could take these factors into account.

### 2.2. Types of Sources

This scoping review included any type of study design.

### 2.3. Search Strategy

The search strategy aimed to locate both published and unpublished studies. An initial limited search of MEDLINE, the Cochrane Library, and JBI Evidence Synthesis was undertaken by one reviewer (MY) to identify articles on the topic. The text words contained in the titles and abstracts of relevant articles, and the index terms used to describe the articles were used to develop a full search strategy (Appendix A). The search strategy was developed with support from a librarian. The databases searched included CINAHL, Embase, Scopus, the Cochrane Register of Control Trials, and Epistemonicus. The search strategy, including all identified keywords and index terms, was adapted for each included database and/or information source. The reference list of all included sources of evidence was screened for additional studies, and forward citation searching was undertaken. Sources of unpublished studies/grey literature included well-known HIV/AIDS-related websites such as WHO, UNAIDS, and PERFAR. Searches were conducted in November 2023. We focused on literature published since 2000 because HIV rapid tests [24] and combined HIV drug therapies [25] began to become widely available in the 2000s. The context and consequences of testing, therefore, altered significantly from the 2000 period onwards.

### 2.4. Study/Source of Evidence Selection

Following the search, all identified citations were collated and uploaded into Covidence (Veritas Health Innovation, Melbourne, Australia), and all duplicates were removed. Following a pilot phase where 10% of titles and abstracts were screened against the inclusion criteria by two independent reviewers (MY, SL), the remainder were screened by one reviewer (MY) for assessment in the review. The full text of selected citations was assessed in detail against the inclusion criteria by MY. Reasons for the exclusion of sources of evidence at the full-text stage that did not meet the inclusion criteria were recorded and reported in the scoping review. The results of the search and the study inclusion process were reported in full in the final scoping review and presented in a PRISMA-ScR [26] flow diagram. Excluded studies can be found in Appendix A.

### 2.5. Data Extraction

Data were extracted from studies included in the scoping review by MY. Data extraction was guided by a modified JBI data extraction tool (Appendix A). The tool was modified during the pilot phase and modifications are detailed within the review. The data extracted included specific details about the participants, concept, context, study methods and key findings relevant to the review questions. A 5-item TIDieR-Lite checklist [27] (Appendix A) was used to map intervention components (By Whom, What, Where, To What Intensity, How Often) to standardised the way in which interventions in the included studies were characterised.

### 2.6. Data Analysis and Presentation

The aim of this review was to map and understand the evidence on voluntary HIV testing and counselling in occupational settings. Hence, the data are presented in narrative and tabular formats to facilitate the identification and summarisation of evidence.

## 3. Results

### 3.1. Study Inclusion

The search identified a total of 6878 records. A further 107 records were obtained from Google Scholar (100 new records) and hand searching reference lists (7 new records), summing to 6985 records. After removing duplicates, 4950 records were screened. During the title and abstract review phase, 4813 records were excluded primarily due to their lack of relevance to HIV testing and counselling in workplaces.

Subsequently, 137 records were selected for full-text review. Of these, 120 articles did not meet the inclusion criteria and were excluded. The reasons for exclusion included non-relevance to workplace VCT (*n* = 39), no HIV-related VCT intervention (*n* = 28), focused on sex workers (*n* = 24), addressed blood fluid exposure (*n* = 6), focused on HIV prevalence in workplaces (*n* = 6), or stigma related to HIV testing (*n* = 5), only evaluated costs of VCT (*n* = 3), involved mandatory (rather than opt-in) VCT (*n* = 3), HIV vaccination related articles (*n* = 2), focused on HIV treatment (*n* = 2), ongoing study protocol (*n* = 1), and VCT service evaluation (*n* = 1).

Finally, 17 articles met inclusion criteria for data charting and summary about HIV VCT in workplace settings. Figure 1 shows the study selection process. A list of excluded studies with reasons can be found in Appendix A.

### 3.2. Characteristics of Included Studies

The studies included in this review reflect the global landscape of workplace HIV-related VCT interventions across diverse populations and occupational settings. The 17 identified reports related to 12 distinct VCT interventions. Studies were conducted in various countries across Europe, North America and sub-Saharan Africa. Studies were undertaken in the UK [14,28,29,30,31,32], Italy [33], the Netherlands [34], South Africa [35,36,37,38,39], Uganda [40], Zimbabwe [41], Nigeria [42], and Canada [43]. The studies were published between 2001 and 2022.

Various study designs were applied to evaluate the interventions, but surveys [28,29,30,31,33,42] and interviews [14,28,29,32,39] were predominantly used. Two studies were pilot trials [40,43], and one study employed a cluster randomised control trial design [41]. The other study designs included prospective cohort study [36], case study [35], retrospective analysis [34], and quasi-experimental design [37,38].

The sample sizes for the included workplace HIV testing and counselling interventions ranged from 17 to 9723, with a total of 20,985 participants. Age ranges spanned from 15 to 70 years. Information about gender was not available in some studies [28,35,41]. However, the remaining studies included a total of 9482 male and 6418 female participants.

The settings of the included studies varied widely, reflecting diverse workplace environments across different sectors and regions. In the UK studies, settings included construction [14,31,32] or mixed settings, including leisure, manufacturing, distribution/retail, healthcare, and food production [28,29]. Three studies were conducted in two South African automotive companies [37,38,39]. Settings for the remaining studies included fishing communities [40], manufacturing of various goods (hardware, construction, industrial, clothing, food), telecommunications [41], an industrial company [35], agricultural migrant workers [33], a rural South African factory [30], a sugar mill community [36], service-based industries [42], a brewing company in sub-Saharan Africa [34], and a hospital in Cape Town, South Africa [43]. The characteristics of the included studies are presented in Table 1.

### 3.3. Characteristics of HIV Testing and Counselling Interventions

The interventions were delivered by healthcare professionals, including doctors, nurses, and sexual health specialists. Different HIV testing products were used, mostly rapid blood tests [14,28,29,31,32,33,34,41] and oral fluid self-tests [40,43]. The interventions primarily consisted of one-time workplace events [14,28,29,31,32,33,37,38,39,41,43]. Most interventions linked participants to care after HIV testing [28,29,35,40,41]; however, only a few studies [34,36,40,43] reported a specific linkage rate to care pathways or processes (patient entry into specialist HIV care after diagnosis), with an average rate of 86.85%. The specific features of the interventions were described using the TIDieR-Lite checklist [27]. Table 2 presents intervention-related information.

The interventions were implemented in organisations of varied sizes, types and sectors. The Healthy Hub Roadshow intervention [28,29] was implemented in 11 organisations, including three small, four medium, and four large-sized companies from various sectors, including leisure, manufacturing, distribution/retail, hospital, local authority, food production, and food industry. Corbett and colleagues [40] implemented intervention in small- and medium-sized businesses (22 in total) involving the manufacturing of hardware, construction, or industrial goods (*n* = 14), clothing (*n* = 3), food (*n* = 3), and telecommunications (*n* = 2). The Test@Work intervention [14,31,32] was hosted at 16 construction sites across 10 participating organisations (over 21 events in total), including one medium- and nine large-sized companies. Additionally, an HIV testing and counselling campaign with incentives was conducted in two medium-sized automotive companies [37,38,39].

On-site HIV test uptake ranged from 23 to 100%, with an average of 63%. Only one study reported off-site uptake (outside of the workplace), and this rate was considerably lower (4.3%) than that observed in the other included studies [41]. Two studies reported gender differences in participation in HIV testing and counselling interventions, with men being less likely to attend than women [30,34].

### 3.4. Facilitators and Barriers to HIV Testing and Counselling Interventions

Overall, the evidence suggests a positive perception and acceptance of HIV-related VCT interventions across the workplace settings among intervention participants [14,29,30,31,33,34,36,41,42,43], employers [28,32,35] and service providers [32,40]. Some studies reported that interventions were effective in reaching previously untested populations [29,31,40,41]. These studies reported high proportions of their participants were first-time testers: 75% [29], 1 in 4 [40], 85% on-site and 87.5% off-site [41], and 78% [31].

Important facilitators of uptake for testing participants were the accessibility of voluntary HIV testing and counselling testing during working hours (i.e., not needing to take time off work) [14,29], with higher uptake of on-site testing compared to off-site testing options [41]. Participants valued the convenience, anonymity, and confidentiality of workplace testing [30], and the inclusion of VCT by embedding it within a general health check [14,28,29,30,32,33] (i.e., normalising testing by combining HIV-related VCT with other health checks and tests, such as obesity, hypertension, cholesterol, and diabetes). Participants also valued free testing [29], and the availability of non-invasive rapid tests that had immediate results [43]. Uptake of testing was facilitated by peer-to-peer support, creating a sense of social cohesion and collective effort, and the use of incentives (such as lottery incentive schemes, free t-shirts or salary prizes) [38,39]. In the brewing sector, workers in sub-Saharan Africa witnessed improvements in the health of HIV-infected colleagues after testing, which encouraged uptake [34]. One study showed that providing education about HIV, condom distribution, and therapeutic options for those who tested positive increased participation in HIV VCT [36].

From the perspective of employers, managers and VCT service providers, facilitators of successful workplace HIV testing and counselling were at organisational level, management level and worker level. A key facilitator was well-organised workplace events (e.g., in terms of event promotion, planning and facilitating through pre-booking appointments [32]). Uptake of workplace VCT was found to be facilitated by peers (other workers), via ‘peer educators’ (e.g., workers having a role in disseminating information, creating strong communication, decreasing stigma, and encouraging participation across different departments and divisions of the organisation [35]) and ‘peer distributors’ of self-test kits [40].

Several barriers were reported to impede the uptake of HIV-related VCT. Stigma and discrimination associated with HIV/AIDS were significant barriers [14,29,35,39]. For example, the UK-based Healthy Hub Roadshow study suggested that limited knowledge and stigma about HIV appeared to be linked to decreased participation in testing [29]. While one study reported attempts to decrease stigma around HIV testing through group discussions at work, it was still highlighted that fear and stigmatisation were barriers [39]. In some settings, employees reported that ‘hyper-masculinity’ in the workplace culture discouraged them from seeking help for sexual and mental health issues [14]. Findings from post-test questionnaire responses reported several barriers to uptake of workplace HIV testing: low perceived risk [29,34,40,42], lack of confidence [29], having already been tested [31,32], and fear of positive test results [14,29]. In the Test@Work studies, both employees and employers reported being busy with work commitments as a barrier to uptake [14,31,32].

A few studies revealed organisational barriers to participation, including perceived lack of support at work [14,35] and differences between groups of workers in terms of awareness or access to health-related support in the workplace. Qualitative data from the Test@Work study suggested that office staff and permanent employees were more likely to be aware of available support or interventions in the workplace than contractors/agency staff, who were more likely to report facing challenges in accessing support from site managers [14]. In a mining company, several challenges were reported, including a lack of policies such as “reasonable accommodations” for HIV-positive workers with inconsistent implementation among departments and the absence of managerial expertise, monitoring, and evaluation [35].

In a ‘Wellness Day’ for factory workers (which included HIV testing), follow-up testing was less well attended by male participants, smokers, and young participants (aged 19–29) [30]. Similarly, Jones and colleagues [31] found that younger participants were more reluctant to participate in health check interventions including HIV testing. Figure 2 and Figure 3 summarise the key facilitators and barriers to HIV testing and counselling in the workplace setting identified in the included studies. Facilitators and barriers to participation in workplace HIV testing and counselling are reported for each study in Table 3.

## 4. Discussion

This is the first scoping review to map out the nature of the global evidence on initiatives and interventions for vocationally active adults on voluntary HIV testing and counselling (VCT) in occupational settings and to summarise facilitators and barriers to the delivery of and/or engagement with VCT initiatives/interventions in the workplace.

Overall, there were 17 identified articles reporting on 12 workplace VCT interventions. Studies were conducted in eight countries, clustering in the African region and Europe (mostly the UK), with one study in North America. There may be myriad reasons why the focus on workplace VCT may be more common in certain regions. However, it could be partially explained by the fact that, despite substantial variation in HIV prevalence across localities [44], the African region has the highest prevalence of HIV globally (an estimated 25.6 million people in 2022 [45] and, therefore, initiatives to increase access to testing are prevalent. Europe and North America have the highest per capita spending on wellness initiatives than other regions of the world [46], and a burgeoning government-level focus on workplace and health, which includes economic arguments for employers to engage with workplace health initiatives [47].

Interventions were delivered in organisations of different types, sizes and sectors. This demonstrates the potential viability of this health testing approach across occupational settings and diverse worker populations. The successful implementation of workplace VCT in a range of occupational settings concurs with prior survey research in which employers reported positive views towards the concept of workplace HIV testing, with many considering offering HIV testing for their workforce in the future [13]. Although the uptake rate for HIV testing varied across the included studies (ranging from 23% to 100%), on average, two-thirds of participants in the interventions received an HIV test on-site at their workplace. In this review, many of the interventions were delivered in a geographical region with a high prevalence of HIV or were delivered in occupational settings through which disproportionately affected populations could be reached (e.g., in our review, these studies included migrant workers and male-dominated industries such as fishing, agriculture, mining, and construction). It is, therefore, possible that, as a community testing route, workplace HIV-related VCT interventions may contribute to reducing inequalities in testing access. This is particularly important since some of the included studies reported that workplace VCT initiatives reached many first-time testers, further supporting the workplace as a potential venue for community testing. Indeed, data from the UK show that testing people for HIV through community services reaches more first-time testers than national self-sampling schemes [47]. Further, prior studies of general workplace health checks have suggested that delivering health interventions through the workplace setting may help to access groups that are considered hard to reach by other routes (e.g., low-paid workers living in socially and economically deprived areas [48]).

However, a key finding from this scoping review is that the exact reach of these workplace interventions across employment settings and worker populations could not be fully determined since most of the studies did not report the characteristics of the organisations in which they were implemented, and some did not provide details about the participating workers. To be able to fully synthesise the published evidence on intervention reach, there is a clear need for more consistency in the description of settings and populations for workplace VCT.

Although there was heterogeneity in the nature of the interventions delivered, all testing was delivered by healthcare professionals (doctors, nurses or sexual health providers), using rapid blood or oral fluid tests. Where reported, the average linkage to care rate in workplace VCT interventions (86.85%) was satisfactory compared to that found in other community HIV testing initiatives (e.g., 89% [49]; 95%, 95% CI = 87–98% [50]) and higher than rates reported for self-testing in the community (e.g., 56% [40]). Most workplace VCT studies, however, did not report linkage to care.

Most of the studies reported evaluations of one-off health events that were either focused on or included VCT. The Test@Work study [14,31,32], for example, embedded HIV testing within a general health check (one-off health event with a range of health tests and checks), and this approach was perceived by organisations, workers, and testing providers to normalise HIV testing and reduce HIV-related stigma. Although views might vary according to setting, workforce gender balance, or cultural norms (Test@Work was conducted in the UK construction industry), the inclusion of HIV testing within a wider package of opt-in health tests and checks was proposed by employers as the most appropriate, if not the only, way to engage the workers in on-site HIV testing.

In terms of the delivery format, it was unclear why one-off events were more common, and there were not enough interventions to meaningfully explore differences between on-off events and longer/repeated interventions (i.e., in reach, uptake, or implementation enablers or barriers). It is possible that one-off events may have been selected by researchers as the limited time input helps them to persuade employers to sign-up as host organisations. Our suggested reasons are twofold; first, one-off events offer an option for workplace health intervention that can be easily slotted into an ongoing programme of workplace health initiatives already being offered by employing organisations. Second, with one-off events, organisations without existing workplace health programmes can experience a ‘taster’ of how interventions could be implemented within the organisation (and how well they are received) without excessive time and effort. The latter may be important to engage small- to medium-sized enterprises (SMEs) in VCT that may have less structured support and resources available to invest in workplace health [51]. Finding ways to engage SMEs in workplace health promotion initiatives is particularly relevant since SMEs are the backbone of economies worldwide, accounting for over 95% of firms and employing 60–70% of the global workforce [52]. Their broad reach makes increasing access to workplace health promotion vital in these settings.

An important finding of this review was that interventions were not consistently described, which made it difficult to make direct comparisons between interventions (e.g., delivery format and intervention type), draw firm conclusions about the appropriateness of workplace VCT for diverse host organisations and recipients (e.g., workplace types, sizes and sectors, geographical regions, worker populations), and reflect on their findings. In this review, we used the TiDiER-Lite checklist [27] to extract intervention details. We recommend that researchers use this checklist (or the full version [27]) to foster consistency in the reporting of future evaluative studies relating to workplace VCT.

Overall, this review identified more factors enabling the uptake of workplace VCT than barriers. A study of routine health check attendance (not specific to VCT or the workplace setting) showed that those least likely to attend routine health checks were men on low incomes, low socio-economic status, unemployed or less well-educated [53]. Workplace-delivered health testing may, therefore, reach populations who may not otherwise access this through other settings. Findings from this review suggest that the uptake of workplace HIV-related VCT was facilitated by the ‘convenience’ of accessing health tests at work (on-site), the ‘immediacy’ of results using rapid tests, the provision of ‘free’ testing and condoms, ‘incentives’ to participate, and the provision of HIV-related ‘education’. The high value placed on the convenience of accessing HIV tests at work aligns directly with findings from prior evaluations of general health checks in workplace settings [12]. Offering HIV-related education alongside HIV testing seems important since limited knowledge about HIV, low risk perceptions and HIV-related stigma were key barriers to workplace VCT uptake in the included studies. Indeed, studies have found that educational intervention can improve men’s behavioural intentions to engage in health screening [54]. Our review findings show that being male, and a masculinity-driven workplace culture, were barriers to uptake of workplace HIV VCT. This concurs with other research showing that men are less likely to attend health screening (including HIV testing), than women, with male-dominant barriers to testing uptake including a heterosexual self-presentation [54].

Factors relating to the organisational context could also be barriers or facilitators of testing uptake. For example, uptake was enhanced when VCT was delivered within well-organised events (e.g., a health check event) which involved the commitment and support of managers and peers for distributing test kits or helping to organise or implement events. Lack of time due to work commitments was a barrier to testing uptake for some. Importantly, there were differences between ease of access for different occupational groups, with more challenges to access experienced by contract workers and agency staff than permanent and office-based staff. Some workers felt there was a lack of support for engaging in health initiatives at work; this could be at the manager level (perceived lack of manager support) or organisational level (e.g., lack of organisational policy surrounding workplace health and clarity around which staff groups could, or could not, access this during working hours).

### 4.1. Limitations of the Review

The review is limited to studies published in the English language and the small number of interventions in this field. The searches were conducted up to November 2023 and, therefore, there may be more reviews published from December 2023 onwards. We found that methods for workplace VCT interventions vary considerably, with many of the studies using surveys or qualitative interviews with stakeholder groups. We did not systematically collect information about how interventions were funded (e.g., by employers, external donors, or as part of research grants), which could be explored in future studies to assess the feasibility of scale-up and long-term sustainability of workplace VCT. This review aimed to map out the nature of the evidence, and it was, therefore, beyond the scope of this review to determine VCT intervention ‘effectiveness’ in terms of diagnostic, clinical or health outcomes, which could be investigated in high-quality randomised controlled trials.

### 4.2. Limitations of Included Studies

Scoping reviews do not include a requirement to assess methodological quality of included studies [19]. Nonetheless, this scoping review highlights that the reporting quality of workplace VCT interventions and their associated evaluations is variable, which means we were unable to meaningfully report comparisons on intervention type, duration, and frequency, or the enablers and barriers to implementation in different occupational settings. This could be examined in future research.

Regarding study designs, we observed that randomised controlled trial (RCT) designs were uncommon among our included studies, with two pilot trials and just one study which reported findings from a full-scale cluster randomised trial. Although RCTs are considered the ’gold standard’ of evaluative health and medical research, it has long been recognised that the RCT may not be the most appropriate design for evaluating occupational health interventions [55]. It has previously been reported that the adoption of RCTs is scarce in the evaluation of workplace occupational health interventions compared to their use in the medical sciences due to the challenges of conducting RCTs in occupational settings [56]. Such challenges may include working with (often changing) gatekeepers, layers of ‘red tape’, competing organisational priorities and workplace policies, data sharing issues, randomisation processes and risk of contamination, and the lack of timeliness of RCTs in generating outcomes of perceived value in uncontrolled, dynamic ‘real-world’ contexts. If conducting an RCT, researchers should ensure that organisational issues are well-considered in the RCT design and consider reporting using a RCT checklist, which takes organisational issues into account [56]. Alternatively, future studies might consider the challenges of undertaking an RCT in employment settings and consider alternative evaluative research designs [55], such as a stepwise approach or a realist evaluation.

### 4.3. Reflexivity

Due to time and resource constraints, no patient or public involvement was undertaken as part of this review. All team members have previously undertaken and published evidence reviews. The project lead is a female health psychologist with expertise in public health and workplace health promotion. The project collaborator is a registered nurse with expertise in HIV and sexual health. Both these team members are female and conducted the Healthy Hub and Test@Work studies, which are included in this review. This may have influenced their interpretation of the review findings. The two researchers (one male, one female) involved in the review data collection and extraction have backgrounds in health research and were not involved in the design or delivery of any of the included studies.

## 5. Conclusions

This scoping review is the first to identify the published evidence for workplace HIV counselling and testing interventions in a global context. Despite the limited number of studies, the workplace appears to be a viable and potentially valuable route to the delivery of community VCT. The uptake rates combined with a high number of enabling factors indicate that such interventions are largely acceptable to workers, employers and service providers. Workplace VCT could, therefore, contribute to improving access to HIV testing and early diagnosis of HIV. However, there are several barriers to participation and organisational challenges that need to be considered. In terms of actionable recommendations, this review suggests that, in the delivery of workplace VCT, we should provide education (to address poor knowledge and low risk perceptions) and make testing easy to access, convenient, and confidential/private. Rapid tests for immediate results are valued. Embedding HIV VCT within general health checks helps to normalise testing and reduce HIV-related stigma. Raising health awareness within organisations and ensuring top-level support for health events is critical, particularly in a masculinity-driven workplace culture. Reporting quality of interventions and associated evaluations is highly variable and could be improved with the use of appropriate checklists to enhance the quality and consistency of descriptions of the characteristics of workers, occupational settings and interventions in workplace VCT. There is a clear need to enhance consistency in study outcomes measured and reported. Further research is warranted to explore differences between intervention types (e.g., one-off events versus longer/repeated interventions) on reach, uptake, acceptability and outcomes. There is scope to further examine differences in reach, uptake, acceptability, and outcomes within and between different worker populations, job roles, work patterns, occupational settings (organisation type, size, sector/industry) and locations.

## Figures and Tables

**Figure 1 ijerph-22-00263-f001:**
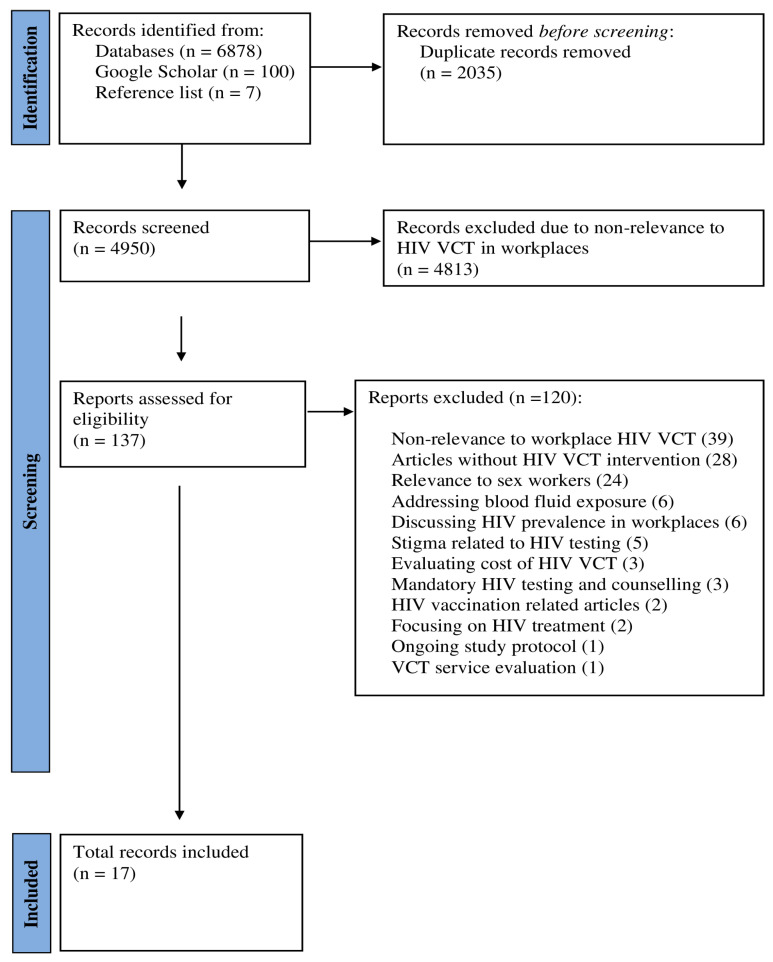
Study selection process.

**Figure 2 ijerph-22-00263-f002:**
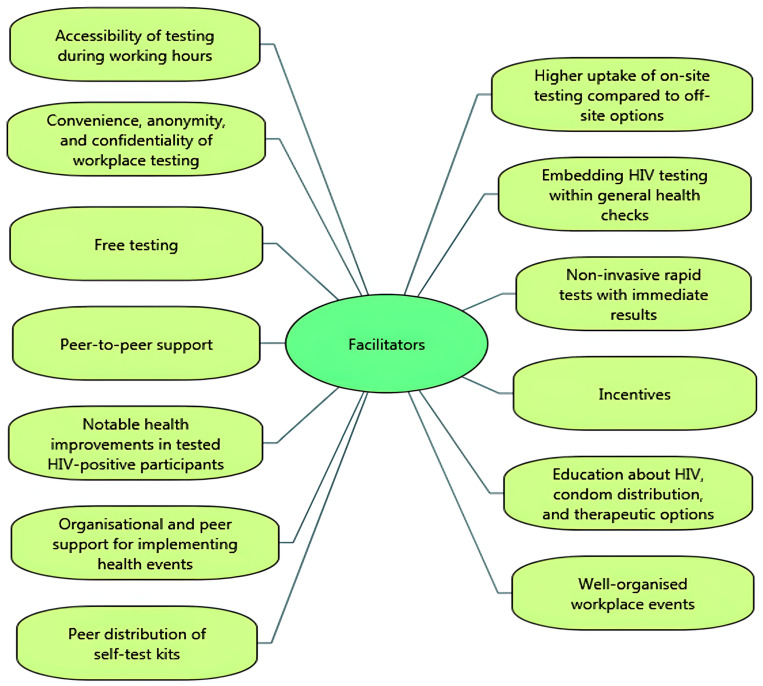
Facilitators of HIV testing and counselling in the workplace setting.

**Figure 3 ijerph-22-00263-f003:**
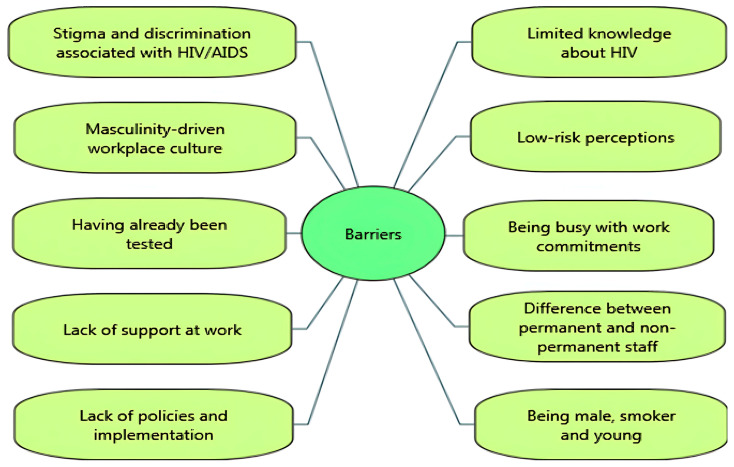
Barriers to HIV testing and counselling in the workplace setting.

**Table 1 ijerph-22-00263-t001:** Characteristics of included studies.

Author Year Country	Title	Aim	Designs	Population	Sample Size	Age Range	Gender	Settings
[28] Blake et al. 2019 UK	Employers’ views of the “Healthy Hub Roadshow”: a workplace HIV testing intervention in England	To evaluate employer uptake and perceptions of workplace HIV testing as part of the Healthy Hub Roadshow, a multi-component general health check intervention in England	Mixed-methods study	Employees from 11 organisations (including migrant workers)	776	N/A	N/A	Various sectors including leisure, manufacturing, distribution/retail, hospital, local authority, food production, and food industry.
[29] Blake et al. 2018 UK	Healthy Hub Roadshow: Employee perceptions of a workplace HIV testing intervention in England	To evaluate the feasibility of delivering HIV testing within workplace health checks, describe the demographic characteristics of employees who chose to attend, and ascertain the views of attending employees towards workplace HIV testing	Mixed-methods study	Employees from 11 organisations (including migrant workers)	776	18–50+	Men: 396 Women: 375	Various sectors including leisure, manufacturing, distribution/retail, hospital, local authority, food production, and food industry.
[40] Choko et al. 2018 Uganda	A pilot trial of the peer-based distribution of HIV self-test kits among fishermen in Bulisa, Uganda	To pilot a peer-based distribution model of HIV self-test (HIVST) kits among fishermen	Single-arm pilot trial	Fishermen	108	18–59	Men: 108	Fishing communities
[41] Corbett et al. 2006 Zimbabwe	Uptake of workplace HIV counselling and testing: A cluster randomised trial in Zimbabwe	To investigate the impact of rapid HIV testing at the workplace on uptake of voluntary counselling and testing (VCT)	Cluster randomised control trial	Workers	2543	N/A	N/A	Manufacturing of hardware, construction, or industrial goods, clothing, food, and telecommunications
[35] Dickinson 2003 South Africa	Managing HIV/AIDS in the South African workplace: just another duty?	To examine the interplay of official policies on HIV/AIDS and the actual practices of HIV/AIDS programmes within a large South African corporation	Case study	Employees	One company (industrial company)	N/A	N/A	Industrial company
[33] Di Gennaro et al. 2021 Italy	Low-wage agricultural migrant workers in Apulian ghettos, Italy: General health conditions assessment and HIV screening	To assess general health conditions and HIV prevalence by giving hygienic and sanitary sustenance	Quantitative survey study and medical assessment.	Agricultural Migrant Workers	321	18–56	Men: 298 Women: 23	Agricultural communities
[30] Houdmont et al. 2013 UK	Acceptance of repeat worksite HIV voluntary counselling and testing in a rural South African factory	To assess the factors that predict repeat VCT attendance at 12-month follow-up	Quantitative survey study	Factory workers	2138	19–50+	Men: 962 Women: 1274	A factory
[31] Jones et al. 2021 UK	Test@work: evaluation of workplace HIV testing for construction workers using the RE-AIM framework	To evaluate Test@Work, a workplace HIV testing intervention for construction workers	Mixed-methods study	Construction workers	426	17–70	Men: 348 Women: 78	Construction sites
[36] Morris et al. 2001 South Africa	A package of care for HIV in the occupational setting in Africa: Results of a pilot intervention	To evaluate a package of care for HIV in the occupational setting in sub-Saharan Africa, specifically in a sugar mill in South Africa.	Prospective cohort study	Sugar mill workers	386	23–62	Men: 386	A sugar mill
[42] Onoja et al. 2020 Nigeria	Voluntary counselling and testing for HIV among allied workers in rural area of igeria: Evaluation of community-based interventions	To evaluate the impact of community-based intervention towards the prevention and control of human immunodeficiency virus on the voluntary testing for human immunodeficiency virus among allied workers in rural Bonny Kingdom of Rivers, State, Nigeria	Quantitative survey study	Workers	587	15–49	Men: 338 Women: 249	Service-based industries including the gas plant and related investments in oil terminals and natural liquid gas production
[34] Van der Borght et al. 2010 Netherlands	Long-term voluntary counselling and testing (VCT) uptake dynamics in a multi-country HIV workplace programme in sub-Saharan Africa	To report the uptake dynamics of voluntary counselling and testing (VCT) during the first 6 years of a HIV workplace programme	Retrospective analysis	Brewing workers	9723	15–45+	Men: 5744 Women: 3865	Heineken’s local operating companies in 14 sites in five sub-Saharan African countries
[43] Pai et al. 2013 Canada	Will an unsupervised aelf-testing strategy for HIV work in health care workers of South Africa? A cross-sectional pilot feasibility study	To assess the feasibility of an unsupervised self-testing strategy for HIV specifically among healthcare workers in South Africa	Pilot cross-sectional study	Healthcare workers	251	18–44	Men: 53 Women: 196	Hospital
[14] Somerset et al. 2021 UK	Accessing voluntary HIV testing in the construction industry: A qualitative analysis of employee interviews from the Test@Work Study	To investigate the experiences of employees with voluntary workplace HIV testing as part of Test@Work general health checks	Qualitative interview study	Construction workers	426	17–70	Men: 348 Women: 78	Construction sites
[32] Somerset et al. 2022 UK	Opt-in HIV testing in construction workplaces: an exploration of its suitability, using the socioecological framework	To explore the suitability of opt-in HIV testing within construction workplaces using a socioecological framework	Mixed-methods study	Construction workers	52	N/A	Men: 26 Women: 26	Construction sites
[37] Weihs et al. 2018 South Africa	The influence of lotteries on employees’ workplace HIV testing behaviour	To understand how lottery incentives influenced the HIV counselling and testing (HCT) behaviour and behaviour intention of shop-floor workers	Post-test-only quasi-experimental study	Shop-floor workers	198	29–40+	Men: 126 Women: 72	Two automotive companies
[38] Weihs and Meyer-Weitz 2016 South Africa	Do employees participate in workplace HIV testing just to win a lottery prize? A quantitative study	To determine whether workers intend to test for HIV only to win a lottery prize	Post-test-only quasi-experimental study	Shop-floor workers	514	29–40+	Men: 341 Women: 173	Two automotive companies
[39] Weihs and Meyer-Weitz 2014 South Africa	A lottery incentive system to facilitate dialogue and social support for workplace HIV counselling and testing: A qualitative inquiry	To explore qualitatively the influence of a lottery incentive system (LIS) as an intervention to influence shop-floor workers’ workplace HIV testing behaviour	Qualitative interview study	Shop-floor workers	17	20–59	Men: 8 Women: 9	Two automotive companies

**Table 2 ijerph-22-00263-t002:** Characteristics of voluntary HIV testing and counselling interventions based on the TIDieR-Lite Checklist.

Intervention	Studies Applying the Intervention	By Whom	Product	Linked to Care	Linkage %	Company Size	Sector	Intensity	Frequency	Uptake %	Gender Difference
Healthy Hub Roadshow	Blake et al. 2019 [28] Blake et al. 2018 [29]	Healthcare professionals	Fourth generation INSTI^®^ finger-prick rapid test	Yes	NR	3 small-sized companies, 4 medium, 4 large	Private and public	One test	One-time event	52	NR
Peer-based HIV test distribution	Choko et al. 2018 [40]	Oral fluid self-test	The OraQuick ADVANCE^®^ Rapid HIV-1/2 Antibody Test	Yes	100	NR	NR	One test	1 month	81.9	Only males
N/A ^a^	Corbett et al. 2006 [41]	Nurses	Determine^TM^ and Unigold^TM^ with either venous or finger-prick blood	Yes	NR	Small and medium sizes (22 in total)	NR	One test	One-time event	51 on-site 4.3 off-site	NR
N/A	Dickinson 2003 [35]	Healthcare professionals	NR ^b^	Yes	NR	NR	NR	NR	NR	NR	NR
N/A	Di Gennaro et al. 2021 [33]	Healthcare professionals	Third generation capillary HIV blood rapid test/Alere Determine^TM^, Abbott	NR	NR	NR	NR	One test	One-time event	100	NR
Wellness Day	Houdmont et al. 2013 [30]	Nurses	NR	Yes	NR	NR	Private	Two tests	Follow-up test after 12 months	84	Men were less likely to attend at follow-up than women
Test@Work	Jones et al. 2021 [31] Somerset et al. 2021 [14] Somerset et al. 2022 [32]	Sexual health professionals	Fourth generation Alere Determine™ HIV-1/2 test kit	Yes	NR	One medium size and nine large sizes	Private	One test	One-time event	81.7	Male: 348 Female: 78
A package of HIV care	Morris et al. 2001 [36]	Healthcare professionals	NR	Yes	82.8	NR	NR	One test	Over a period of one year, from 1999 to 2000	26.4	Only males
N/A	Onoja et al. 2020 [42]	Healthcare professionals	NR	Yes	NR	NR	NR	NR	Provided for 3 years	68	NR
Heineken HIV workplace programme	Van der Borght et al. 2010 [34]	Healthcare professionals	Blood rapid test	Yes	64.6	NR	Private	Three tests	Every 2 years from 2001 to 2007	23	Men: 22 Women: 28
N/A	Pai et al. 2013 [43]	Oral fluid self-test	The OraQuick ADVANCE^®^ Rapid HIV-1/2 Antibody Test	Yes	100	NR	Public	One test	One-time event	99.2	NR
A lottery incentive system (the LIS)	Weihs et al. 2018 [37] Weihs and Meyer-Weitz 2016 [38] Weihs and Meyer-Weitz 2014 [39]	Nurses	NR	NR	NR	Two mid-sized automotive companies	Private	One test	One-time event as a part of the HCT campaign	NR	NR

^a^ Where N/A is listed in the intervention column, this denotes an absence of intervention ‘name’ with HIV testing and counselling being a workplace provision (whether one-off or ongoing). ^b^ NR = Not reported.

**Table 3 ijerph-22-00263-t003:** Facilitators and barriers to participation in workplace HIV testing and counselling in included studies.

Study	Facilitators	Barriers	Key Outcomes
[28] Blake et al. 2019	⇒ The inclusion of HIV testing as part of a wider health check ⇒ The perceived trustworthiness of delivery partners, the ability to provide engaging opportunities for employee health ⇒ Having visible top-level managerial support	⇒ Having limited budgets for future events ⇒ Concerns about loss of employee productivity related to attendance for testing during work time ⇒ Lack of support and guidance around HIV testing at work ⇒ Management opinion on HIV testing as an inappropriate service to offer employees	⇒ This study highlighted the overwhelmingly positive perceptions of employers towards opt-in HIV testing within a general health check.
[29] Blake et al. 2018	⇒ The convenience of having health checks at the workplace with no cost ⇒ Rapid HIV test implementation and immediate test results ⇒ Embedding HIV testing within a wider package of health tests ⇒ Delivery of health checks by an external organisation for the confidentiality of test data ⇒ Personalised advice and feedback from the health checks	⇒ Fear of being HIV-positive and losing job ⇒ Lack of perceived risk for HIV	⇒ The primary findings of the study revealed that of the 776 employees who attended the health check events, 52% opted for an HIV test, with 75% being first-time testers. ⇒ Higher rates of HIV testing were observed in migrant groups, with HIV testing undertaken by 64%. ⇒ The intervention was well-received, with 96% of attendees deeming HIV testing as an acceptable element of workplace health checks. ⇒ Additionally, 79% reported gaining new health knowledge, and 60% chose to receive follow-up health text messages.
[40] Choko et al. 2018	⇒ As the study demonstrates high uptake and accessibility of HIV testing, peer-based self-kit distribution could be considered facilitative	⇒ The reasons for refusals among the individuals to whom HIVST kits were offered included having recently tested, not having taken risks, not being interested, holding intervention about HIVST, and having health issues	⇒ The study’s findings highlight the potential of the peer-based distribution of HIV self-test kits in effectively reaching men who have not previously tested for HIV. With a focus on fishing communities in Uganda, the approach showed high uptake and acceptability, with no reported coercion, and successful linkage to confirmatory testing and treatment for those diagnosed with HIV.
[41] Corbett et al. 2006	⇒ On-site HIV testing linked to basic HIV care ⇒ Being single ⇒ Aged below 25 for both mechanisms of testing, and over 45 for only on-site testing ⇒ Having had past household exposure to TB ⇒ Poorer self-rated health ⇒ Having manual job (not described)	N/A	⇒ The study showed that the uptake of HIV testing was significantly higher in the on-site testing group compared to the off-site voucher group. ⇒ The on-site approach reached a mean uptake of 51.1% across the businesses, while the off-site voucher uptake was only 19.2%. ⇒ Only 4.3% reported using their voucher for off-site testing.
[35] Dickinson 2003	⇒ Peer educators and a medical sister who organised a network of peer educators across all divisions and business units	⇒ Stigma, discrimination, and the complex nature of managing the disease ⇒ Lack of supportive policies such as ‘reasonable accommodation’ for HIV-positive individuals ⇒ Inconsistent implementation ⇒ Lack of managerial expertise, monitoring and evaluation	⇒ This study reveals that Deco’s response was initially fragmented, with various divisions implementing policies independently. These included promotion of voluntary testing, counselling, and openness about the disease. However, the lack of a coordinated strategy and support from senior management led to inconsistent results across the company.
[33] Di Gennaro et al. 2021	N/A	N/A	⇒ All participants were tested for HIV. One participant was found to be HIV-positive.
[30] Houdmont et al. 2013	⇒ The study suggests workplace VCT might be attractive due to convenience (during work hours), anonymity, and confidentiality of worksite provision	⇒ Gender, being male ⇒ Being a smoker ⇒ Being young	⇒ The study reveals that receiving a positive HIV diagnosis, being male, and smoking are all factors associated with a lower likelihood of attending follow-up HIV voluntary counselling and testing (VCT) sessions in a workplace setting. ⇒ In terms of demographic characteristics, men were less likely to attend at follow-up than women.
[31] Jones et al. 2021	⇒ Positive worker and manager feedback about the testing event ⇒ Effective and well-planned event organisation	⇒ Workload ⇒ Having had a health check elsewhere ⇒ Perception of being too young for the health check ⇒ Privacy concerns	⇒ The uptake of HIV testing was substantial, with 97% of health check attendees agreeing to sexual health consultations and 82% opting for HIV testing. ⇒ 78% of those tested had not been previously tested for HIV, and all tests returned non-reactive results. ⇒ Most participants found the testing acceptable and reported gaining new health insights from the event, with many expressing intentions to change health behaviours confidently.
[36] Morris et al. 2001	⇒ Linking a therapeutic option to the services (condom distribution, education about HIV and STIs, enhanced VCT) as a stimulus to uptake.	N/A	⇒ The package of care was successfully implemented and made sustainable, demonstrating measures of behaviour change, such as increased condom usage and a decrease in the number of sexually transmitted infections treated.
[42] Onoja et al. 2020	⇒ Being well-educated ⇒ To reduce fear and anxiety ⇒ To know HIV status	⇒ Poor perception of VCT ⇒ A feeling of not being at risk ⇒ Already been tested	⇒ The pre-intervention survey showed that while knowledge of VCT was at 76.8%, the actual testing rate was only 37.5%. ⇒ Post-intervention results were more positive, with knowledge increasing to 88.9% and testing rates rising to 68.0%. ⇒ The study concludes that community-based interventions can significantly impact the prevention and control of HIV in rural areas by increasing both awareness and uptake of VCT services.
[34] Van der Borght et al. 2010	⇒ Educational campaigns, health education, and information campaigns ⇒ Confidential In-House VCT ⇒ Special events, including World AIDS Day, company family days, and other targeted health campaigns ⇒ Individuals with symptoms or perceived high HIV risk ⇒ Witnessing benefits of treatment in the health of co-workers	⇒ Scepticism about the confidentiality of test results ⇒ Lack of perceived need for testing	⇒ The study found that the annual average VCT uptake among eligible individuals ranged from 15% to 32%, with higher testing coverage among female employees and spouses compared to their male counterparts.
[43] Pai et al. 2013	⇒ Ease of use ⇒ Non-invasive and painless nature of oral testing ⇒ Privacy	N/A	⇒ Over 99% of participants successfully completed the self-testing process, demonstrating the approach’s feasibility. ⇒ All participants who tested positive (100%) were linked to confirmatory testing and treatment within 24 h, highlighting a successful linkage system. ⇒ Over 91% of participants reported a positive experience with the unsupervised self-testing strategy.
[14] Somerset et al. 2021	⇒ Convenient and easy access to health checks in the workplace. ⇒ Rapid testing and personalised feedback and support ⇒ Private and confidential HIV testing settings	⇒ High workloads and long working hours ⇒ Concerns about job security ⇒ Masculinity workplace culture discouraging employees from seeking help for sexual and mental health issues ⇒ Perceived lack of organisational support	⇒ The study demonstrated that HIV testing, delivered in the context of a general health check, is highly acceptable to employees in the male-dominated construction sector and reached individuals who had never had an HIV test, as well as repeat testers.
[32] Somerset et al. 2022	⇒ Peer-to-Peer encouragement ⇒ Accessibility of on-site testing ⇒ Education about HIV testing to an untested population ⇒ HIV testing promoted as part of general health checks ⇒ Use of a finger-prick test instead of a venous blood sample	⇒ Being reluctant to seek healthcare, particularly for sensitive issues like sexual health ⇒ A lack of time due to work demands ⇒ Having already had health checks elsewhere ⇒ Lack of suitable space and facilities	⇒ The research findings indicated that the largely male construction workforce showed high engagement with workplace HIV testing, which was facilitated by peer-to-peer encouragement and appreciated for its accessibility. ⇒ Interviewees from all groups commented on the reluctance of men to seek healthcare, particularly their reluctance to discuss sensitive issues. ⇒ Despite some challenges in planning and providing private facilities, managers recognised the value of offering health checks, and health professionals valued the opportunity to reach an untested population with poor understanding of their HIV risk.
[37] Weihs et al. 2018	⇒ The announcement of lottery incentives significantly influenced subjective norms, boosting employees’ perception that important individuals or groups would approve HIV testing in the company and support them in going for HIV testing. ⇒ Confidence in seeking HCT in the workplace was a significant factor influencing behaviour intention, as employees perceived behavioural control towards workplace HCT behaviour played a role in their intention to participate in HIV testing.	N/A	⇒ The results of this study suggest that the lottery incentives had an impact on the experimental group’s intention to test for HIV and shed some light on the components of the Theory of Planned Behaviour (TPB) model that played a significant role in the prediction of behaviour intention of the experimental group.
[38] Weihs and Meyer-Weitz 2016	⇒ Lottery incentives were a facilitating factor in engaging non-permanent workers with the intervention.	N/A	⇒ In the case of permanent workers, no significant association was found between the behaviour intentions to test in the two settings. For these workers, testing intention was not significantly influenced by their interest in winning a lottery prize. However, for the non-permanent workers, a significant yet small difference was found. When lotteries were announced, non-permanent employees’ HCT behaviour intention was slightly higher in setting 1 than in setting 2, suggesting that some were likely to have participated in HIV testing for entry into the lottery in the hope of winning a prize.
[39] Weihs and Meyer-Weitz 2014	⇒ Excitement and anticipation: The communication of the lottery prizes, date of prize-giving, and entry conditions in advance generated excitement and anticipation among the shop-floor workers, motivating them to participate in workplace HIV testing. ⇒ Social cohesion: The LIS intervention created social cohesion among the workers, as they discussed the prizes and HIV testing openly and supported each other to participate in the testing. ⇒ Group encouragement and peer support: The HCT campaign in the context of the LIS was transformed into a group ‘project’ where mutual encouragement and strong peer pressure to test played a role in the uptake of HCT. ⇒ Open communication: Open discussions and solicitation of group support for HIV testing, as well as communication with families and partners about the LIS, played a crucial role in promoting the uptake of workplace HCT	⇒ Fear of stigmatisation and discrimination ⇒ Perceived lack of confidentiality	⇒ The findings highlight the role of the LIS in transforming workplace HCT into a group project, encouraging mutual support and strong peer pressure to facilitate HIV testing behaviour. ⇒ The study reveals that the LIS not only influenced workplace HCT behaviour but also facilitated open communication and group-based decision-making around HIV testing and lottery incentives.

## Data Availability

The authors confirm that the data supporting the findings of the study are available within the article and its Appendix A.

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
