# Peer review of "Voluntary HIV Testing and Counselling Initiatives in Occupational Settings: A Scoping Review"

_ijerph, 2025, doi:10.3390/ijerph22020263_

Round 1
Reviewer 1 Report
Comments and Suggestions for Authors
The paper is well written; the methodology well-articulated, the introduction set the scene and state the rationale for the review; the results are well presented and comprehensively supported in the discussion section. I enjoyed reading the paper.
Abstract
Mention the Facilitators and barriers to VCT participation on the abstract.
Methods
Types of sources
This scoping review included any type of study design. Please provide justification for this?
Results section
Table 1 - please add a row for country, indicate where each study was done
Table 1- Blake et al, 2018 under "Designs", questionnaires and interviews is not a study design; these are data collection tools. Please also correct for Blake et al 2019, Gennaro et al. 2021, Onoja et al, Somerset et al., 2021, Weihs and Meyer-Weitz 2014. If you also want to indicate the data collection tools, mention that this is "Designs and data collection tools" and report both as such.
It would be great to know if these interventions were funded/supported by the companies/employer or external donors or campaigns to see the sustainability thereof.
Reviewer 2 Report
Comments and Suggestions for Authors
Dear Authors,
I sincerely thank you for the opportunity to review this manuscript, that I read with great interest. I personally think that this work represents a scientific novelty, is well written and structured and presents a solid methodology, I just have some suggestions in order to further improve its quality:
- Lines 56-58: I suggest to add a quoted reference to the U=U concept and to the consequently non-existent risk to transmit HIV in a situation of stable virosuppression, since it can significantly improve this initial introduction to ART efficacy.
- Line 65: I suggest to extensively write "voluntary HIV testing and counselling" the first time you mention it in the body of the review;
- Lines 81 and 145-146: did the Authors perform other searches in literature between 2023 and 2025, to assess the possible publication of recent reviews?
- Line 483: there is a little typo, since the considered language is not mentioned.
In conclusion, I express my congratulations to the Authors for the quality of the manuscript they presented. I personally think that this work can be accepted after minor revisions.
I remain at your disposal.
Best Regards.
Reviewer 3 Report
Comments and Suggestions for Authors
Authors can work on the suggestions provided in the attached file

Quality of English language and flow of text could be improved as suggested
